# The Telomeric Repeats of HHV-6A Do Not Determine the Chromosome into Which the Virus Is Integrated

**DOI:** 10.3390/genes14020521

**Published:** 2023-02-18

**Authors:** Aleksey V. Kusakin, Olga V. Goleva, Lavrentii G. Danilov, Andrey V. Krylov, Victoria V. Tsay, Roman S. Kalinin, Natalia S. Tian, Yuri A. Eismont, Anna L. Mukomolova, Alexei B. Chukhlovin, Aleksey S. Komissarov, Oleg S. Glotov

**Affiliations:** 1Pediatric Research and Clinical Center for Infectious Diseases, 197022 St. Petersburg, Russia; 2SCAMT Institute, ITMO University, 191002 St. Petersburg, Russia; 3Department of Genetics and Biotechnology, Saint-Petersburg State University, Universitetskaya Nab. 7/9, 199034 St. Petersburg, Russia; 4R.M.Gorbacheva Memorial Institute of Oncology, Hematology and Transplantation, Pavlov First Saint Petersburg State Medical University, 197022 St. Petersburg, Russia; 5D.O. Ott Research Institute of Obstetrics, Gynaecology, and Reproductology, 199034 St. Petersburg, Russia

**Keywords:** HHV-6A, ciHHV-6A, chromosomal integration, telomeric repeats, TMR

## Abstract

Human herpes virus 6A (HHV-6A) is able to integrate into the telomeric and subtelomeric regions of human chromosomes representing chromosomally integrated HHV-6A (ciHHV-6A). The integration starts from the right direct repeat (DR_R_) region. It has been shown experimentally that perfect telomeric repeats (pTMR) in the DR_R_ region are required for the integration, while the absence of the imperfect telomeric repeats (impTMR) only slightly reduces the frequency of HHV-6 integration cases. The aim of this study was to determine whether telomeric repeats within DR_R_ may define the chromosome into which the HHV-6A integrates. We analysed 66 HHV-6A genomes obtained from public databases. Insertion and deletion patterns of DR_R_ regions were examined. We also compared TMR within the herpes virus DR_R_ and human chromosome sequences retrieved from the Telomere-to-Telomere consortium. Our results show that telomeric repeats in DR_R_ in circulating and ciHHV-6A have an affinity for all human chromosomes studied and thus do not define a chromosome for integration.

## 1. Introduction

Human beta herpes viruses type 6A (HHV-6A) and HHV-6B belong to the genus Roseolovirus, subfamily Betaherpes virinae, which is widespread in the human population. More is known about the epidemiology of HHV-6B. For example, more than 90% of the human population is infected with it within the first three years of life [1]. HHV-6A is less studied since the pathogen is acquired later in life, is asymptomatic and is more often detected in immunocompromised individuals [2]. HHV-6A/B are mainly transmitted by direct contact e.g., through saliva, and less frequently via droplets, sexual intercourse, blood, and organ transplants.

The human herpes virus 6A genome consists of double-stranded DNA with an average length of 160 kbp. Most of the genes are located in a unique region (U), which is flanked by left and right direct repeats (DR_L_ and DR_R_, respectively) regions. The DRs are in turn surrounded by pac1 and pac2, which are cis-acting signals for the virus packaging [3,4]. Next to pac1 and pac2 in DR are hexanucleotide telomeric repeats (TMR), which are identical to human telomere sequences. The 5’-end of the DR contains imperfect telomeric repeats (impTMR) and the 3’-end contains perfect telomeric repeats (pTMRs). The perfect telomeric repeats consist of conserved hexanucleotide sequences (TTAGGG)n that can be repeated from 15 to over 180 times in HHV-6A [5]. Meanwhile, the impTMR is composed of telomere-like sequences.

The ability of HHV-6A and 6B to integrate into the telomeric or subtelomeric regions of human chromosomes was discovered in 1993 by Luppi et al. [6]. Integration of the virus can occur in 1q, 6q, 9q, 10q, 11p, 17p, 18p, 19q, 22q, and Xp chromosomes, although the integration mechanisms are not fully understood [7,8,9,10,11]. This form of virus is called chromosomally integrated HHV-6A/B (ciHHV-6A/B). Once HHV-6 is integrated into the genome of germ cells, it can be transmitted to subsequent offspring according to Mendelian laws, representing an inherited chromosomally integrated form of HHV-6A/B (iciHHV-6A/B). The prevalence of ciHHV-6A/B carriers varies from 0.6% in Japan and Canada, to 1–3% in the European population, depending on regional and demographic factors [9,12,13].

Cases of telomeric integration of the viral genome into the host chromosomes have been described for six herpes viruses. These are human herpes virus 6A (HHV-6A), human herpes virus 6B (HHV-6B), human herpes virus 7 (HHV-7), Marek’s disease virus (MDV), gallid herpes virus 3 (GaHV-3), and meleagrid herpes virus 1 (MeHV-1) [14].

Marek’s disease virus is the best studied of these herpes viruses. It has been shown to be associated with development of lymphoma in chickens [15]. Similar to HHV-6A, MDV chromosomal integration involves the TMR sequence. Furthermore, deletion of this sequence prevented the virus integration into telomeric regions of chicken chromosomes, leading to a significant reduction in lymphoma development. Meanwhile, the malignant transformation of cells is not associated with the fact of integration itself, but with the expression of several viral proteins and miRNAs [15]. It remains an open question whether such pathogenic and oncogenic features are present in HHV-6A. The literature currently provides data on the possible role of ciHHV-6A/B in the development of lymphoma [16]. A high prevalence of ciHHV-6B infection in the myocardium of patients with heart failure and suspected myocarditis or dilated cardiomyopathy has been reported [17]. Thus, the association of ciHHV-6A/B with the development of various pathologies is a current issue in medicine.

Analysis of ciHHV-6A sequences showed that virus integration proceeds in a specific orientation, starting from the DR_R_ region [18]. Experiments showed that pTMRs are required for integration, whereas the absence of impTMR only slightly reduced the frequency of HHV-6 integration into human chromosomes [19]. Thus, pTMR in the DR_R_ region is critical for chromosomal integration.

A study by Aswad et al. investigating the evolutionary history of HHV-6A and 6B produced interesting results [20]. On the phylogenetic tree, circulating and integrated HHV-6B are intermingled, indicating active processes of integration, release, and reintegration of this virus. At the same time, on the phylogenetic tree with HHV-6A, circulating and integrated viruses diverged into 2 separate branches. In addition, chromosomally integrated HHV-6A subdivided into separate clades according to the site of integration. The ciHHV-6A genomes formed clades of viruses integrating only into 17 (17p), only into 18 (18q), and only into 19 (19q) chromosomes. Viral selectivity at the integration locus could explain this phenomenon. In other words, if there is a factor (or combination of factors) in the HHV-6A genome that determines the chromosome for virus integration.

Since there is a divergence for ciHHV-6A from circulating HHV-6A, Aswad et al. suggest that circulating HHV-6A has lost its ability for chromosomal integration and at the same time integrated HHV-6A does not contribute to the pool of circulating HHV-6A [20]. However, with a larger sample size, this hypothesis may be revised in the future. Another way to test this assumption is to analyse the telomeric repeats of HHV-6A within the DR_R_. If the chromosomal integration starts from the pTMR, it is possible that this region is a factor in determining the integration site.

Thus, the main aim of this work was to determine whether the chromosomal integration site is defined by the telomeric repeat sequence or other tandem repeats in the DR_R_ region of HHV-6A. If the association was found, the DR_R_ sequence of circulating HHV-6 could be used to predict its possible integration into a particular chromosome and to assess the impact on the development of subsequent pathologies. In the absence of this relationship, the hypothesis proposed by Aswad et al. that there are no new cases of HHV-6A integration can be confirmed. The ciHHV-6A separation into clades is due to the presence of a common ancestor.

## 2. Materials and Methods

### 2.1. Sequence Search

HHV-6A sequences and their features were searched using articles [20,21,22,23,24,25,26,27,28,29,30] and the Bacterial and Viral Bioinformatics Resource Center (BV-BRC) open database [31,32,33]. All genomes longer than 130 kbp were included. The NCBI IDs of all sequences collected are available in Appendix A.

### 2.2. Alignment

Global genome alignment was performed using mafft v7.505 [34] for each group separately (circulating viruses, 17p, 18q, and 19q). Additional parameters “–maxiterate 1000” and “–maxambiguous 0.05” were used to set the number of iterations to 1000 and to filter out sequences with a percentage of ambiguous nucleotides (N, any nucleotide) higher than 5%. The GS strain genome (GeneBank: KJ123690) was used as a reference. The classification of viral genomes into clades is given in Table 1.

### 2.3. Identification and Visualisation of Indels

Insertions, deletions, and SNPs were extracted from the alignments using the bio tool (v1.4.0) with the parameter “format–vcf” to obtain files in VCF format [35]. SNPs were not used in further work. The plotly library (v5.5.0) [36] for Python 3 was used for visualisation.

### 2.4. Search for Tandem Repeats

The Tandem Repeats Finder tool (TRF v4.09 [37]) was used to search for telomeric and other tandem repeats. We used the following search parameters: 7 for a match weight, 7 for mismatch penalty, 7 for indel penalty, 80 for match probability, 10 for indel probability, and 50 for minimum alignment score to report. The search was performed for both circulating and chromosomally integrated HHV-6A sequences.

The BLASTN tool from BLAST+ (v2.13.0) was used to detect matches between the tandem repeats found and human chromosomes [38]. BLASTN was run on a custom database generated from human chromosomes 17, 18, 19, and 22, obtained from the Telomere-to-Telomere (T2T) consortium [39]. The GeneBank IDs of the chromosome sequences are CP068267, CP068261, CP068260, CP068259, and CP068256, respectively. In addition, the parameters “-word_size 6” and “-dust no” were specified for BLASTN. The hits were filtered for “sequence identity” > 90% and “e-value” < $1\times 10^{-3}$. The results were then converted into BED format (Appendix A).

## 3. Results

### 3.1. Search Results for HHV-6A Sequences

The literature search yielded 62 HHV-6 genomes, and 4 more were found in the Bacterial and Viral Bioinformatics Resource Center (BV-BRC) database. A total of 66 HHV-6A genomes were collected, together with information on the country and year of isolation, as well as the type of virus (circulating/integrated). Of all the genomes collected, 13 correspond to circulating HHV-6A and the remaining 53 are chromosomally integrated HHV-6A. Among the 53 ciHHV-6A viruses, integration into the 10th chromosome was detected in 1 virus, into the 17th chromosome in 7 viruses, into the 18th chromosome in 2 viruses, and finally into the 19th chromosome in 2 viruses. A table of the genomes used in this study can be found in Appendix A.

In Aswad et al., phylogenetic analysis of 51 HHV-6A genomes showed a division of circulating and integrated viruses in 2 separate branches [20]. In turn, ciHHV-6A is divided into 3 clades in a manner that means, hypothetically, viruses from the same clade integrate into the same chromosome. The first clade corresponds to ciHHV-6A with an integration site in the short arm of chromosome 17 (17p), the second clade includes ciHHV-6A integrated into the long arm of chromosome 19 (19q), and finally the third clade includes viruses that are integrated into the long arm of chromosome 18 (18q). This allowed us to extend our samples as follows: “17p” group to 18 genomes, “18q” group to 15 genomes, and “19q” group to 10 genomes.

### 3.2. Patterns of Insertions and Deletions in the DR_R_

First, to look for any patterns that might distinguish circulating viruses from integrated viruses, we decided to analyse insertions and deletions (indels) in both types of virus. Indels are completely or generally irreversible mutations that can affect intraspecific diversity and shape genes and genomes. Unlike SNPs, indels can affect sites of greater range and are therefore more likely to cause phenotypic changes [40].

According to Wallaschek et al., absence of the left DR region only slightly reduced the number of HHV-6 integrations and was therefore not required for this process. In contrast, in absence of the right DR, there were no cases of virus integration [19]. Therefore, in this study, we focused on the search for indels only in DR_R_ (Figure 1).

The DR_R_ contains six open reading frames (ORFs) (RJ1, DR1-R, DR2-R, DR3-R, DR4-R, and DR6-R) and regulatory sequences. The regulatory sequences include telomeric repeats, pac1 and pac2. The pac1 and pac2 sequences are called due to their role in packaging and cleavage of the viral genome. The telomeric repeats are involved in the HHV-6A integration into human chromosomes. Deletions and insertions in these regulatory sequences could hypothetically affect the functioning of the virus.

To find possible patterns of indel distribution in integrated viruses, we independently aligned circulating viruses and viruses integrating in 17p, 18q, and 19q chromosomes independently to the GS strain (GeneBank: KJ123690.1) which is circulating HHV-6A.

We then filtered out all genomes that did not have the right DR sequence. In these genomes, the DR_R_ was either not sequenced or not fully assembled. The alignment results of the remaining 38 genomes are shown in Figure 2.

Figure 2 shows two distinct deletions at ∼153.5 kbp and ∼154 kbp in almost all chromosomally integrated HHV-6A. The only exception is the KY315558 genome in the “19q” group, which lacks one of the deletions (Figure 2d). This pattern in circulating HHV-6A is only observed in one genome, that of NC_001664, which is one of the HHV-6A reference genomes. It was isolated in Uganda in 1994. This does not allow us to say that these two deletions are specific to ciHHV-6A alone.

In the ciHHV-6A group of chromosome 18 (Figure 2b) there is an insertion at approximately 156.5 kbp. It should be noted that this region in the genome of KY31555 (Figure 2b) is partially located in a region that has not been fully sequenced (shown in grey). In other ciHHV-6A, this insertion is absent (Figure 2c,d). Furthermore, it is not possible to say that the ciHHV-6A group from chromosome 19 (Figure 2d) lacks this insertion because there are not enough viruses for a valid result. Therefore, we can only make a putative distinction between ciHHV-6A from the 18q and 17p integration sites.

Analysis of the data obtained showed that there are no patterns of insertion and deletion distribution that would be characteristic of only chromosomally integrated HHV-6A. There is probably a difference between ciHHV-6A integrated into 17p and ciHHV-6A integrated into 18q. This difference involves an insertion in the region at about 156.5 kbp in ciHHV-6A integrated into 18q. However, more samples of ciHHV-6A from different integration sites and different geographical regions are needed to draw more robust conclusions.

### 3.3. TMR Does Not Identify the Chromosome for HHV-6A Integration

Another approach to detect differences in the DR_R_ of ciHHV-6A integrated into distinct chromosomes is to analyse these repeats directly. Both the left and right direct repeats of HHV-6A contain perfect and imperfect telomeric repeats (pTMR and impTMR, respectively). These are similar to human telomeric repeats. The perfect telomeric repeats consist of conserved hexanucleotide tandem repeats (TTAGGG)n, which can be repeated from 15 to over 180 times in HHV-6A [5]. The impTMR is composed of telomere-like sequences.

Since the integration process starts with the right direct repeats, the analysis focused on their sequences. We first filtered out genomes that had no DR_R_ region or had an ambiguous nucleotide content (N, any nucleotide) of more than 10%. As a result, only 19 out of 66 genomes remained. Their group distribution is as follows: circulating HHV-6A-13 genomes; “17p” group—2 genomes; “18q” group—4 genomes (Table 2).

TMRs and other tandem repeats were searched separately for each genome using the TRF tool. BLASTN was then used to search for these repeats in a custom database including the sequences of 17, 18, 19, and 22 human chromosomes obtained from the T2T Consortium. In other words, these sequences contain completely assembled human telomeric sequences. The 22nd chromosome was added because it is the most common integration site described for the Japanese population [12]. The matches found were filtered for “sequence identity” > 90% and “e-value” < $1\times 10^{-3}$. The results are presented in BED format in Appendix A separately for each of the chromosomes. The results of the local alignment performed by BLASTN can be seen in these tables. The first column denotes the human chromosome at which the alignment was performed. The second and third columns show the coordinates of the start and end (respectively) of the match found. These coordinates are given for the human chromosome indicated in the first column. The fourth column contains the name of the sequence for which a match was found. This name consists of the NCBI ID, the indication of the virus group (circulating, “17p” or “18q”), the country and the number of tandem repeats found by the TRF tool. The last fifth column shows the orientation of the strand of the human chromosome.

Analysis of the tabular data revealed that telomere repeat sequences in ciHHV-6A (from chromosomes 17 and 18), as well as circulating HHV-6A, were found in all human chromosomes involved in this study (chromosomes 17, 18, 19, and 22).

Based on the results obtained, we can conclude that telomeric repeats within the DR_R_ of both circulating and ciHHV-6A have an affinity for all chromosomes tested. This suggests that they do not affect the integration site, while the separation of the clades in the phylogenetic tree is due to the common origin of the viruses.

## 4. Discussion

In this study, we attempted to answer the question of whether the HHV-6A integration site can be determined from the sequence of its telomeric repeats. The assumption of a predetermined integration site follows from the results of Aswad et al. [20]. Phylogenetic analysis of HHV-6A by this research team showed a separation of chromosomally integrated HHV-6A (ciHHV-6A) into three clades, such that viruses from one clade are integrated into the same chromosome (human chromosomes 17, 18 and 19). We hypothesised that the chromosome into which HHV-6A integrates depends on the sequence of perfect telomeric repeats (pTMR) within the right direct repeat (DR_R_) region. This hypothesis is based on two findings. The first is that sequence analysis of inherited ciHHV-6A by Aimola et al. showed that integration starts from the DR_R_ [18]. This is carried out by one of the putative mechanisms of homologous recombination, either break-induced replication (BIR) or single-strand annealing (SSA). Secondly, Wallaschek et al. specify the region required for integration into pTMRs [19]. A similar result was obtained for Marek’s disease virus (MDV), which affects chickens. Deletion of TMR sequences in MDV prevented virus integration into telomeric regions of chicken chromosomes [15]. Based on these facts, we selected telomeric repeats within the DR_R_ to confirm or reject the hypothesis of a pre-determined integration site for each specific chromosome.

We started by analysing insertions and deletions (indels) in the right direct repeat region, as these types of mutations can affect large segments of the genome and are likely to affect the regulatory functions of this region [40]. Indels were analysed separately for circulating HHV-6A, ciHHV-6A from chromosome 17, ciHHV-6A from chromosome 18, and ciHHV-6A from chromosome 19. Indels were detected by aligning each of the virus groups to the reference GS strain genome (GeneBank: KJ123690) which is circulating HHV-6A. Analysis of the distribution of indels in the right direct repeat region showed no differences between circulating and chromosomally integrated HHV-6A (Figure 2). However, we detected an insertion at position 156.5 kbp in a group of ciHHV-6A integrated into 18q. This insertion is not present in other ciHHV-6A, but for more accurate results a further study with more samples of ciHHV-6A from different integration sites and different geographical regions is needed. Interestingly, the insertion in this region is also present in some circulating HHV-6A. The reason for this is not known and may be the subject of further research.

Next, we directly analysed the telomeric repeat sequences (TMRs) within the DR_R_. We first extracted these sequences from the viruses that passed a quality filter. After filtering, 13 circulating HHV-6A genomes, 2 HHV-6A genomes integrated into chromosome 17, and 3 HHV-6A genomes integrated into chromosome 18 remained (Table 2). We performed a local alignment of the TMRs of these viruses on human chromosomes 17, 18, 19, and 22, obtained by the Telomere-to-Telomere consortium [39]. Chromosomes 17, 18, and 19 were chosen because the ciHHV-6As we collected are integrated into these chromosomes, and chromosome 22 was added because it is the most common HHV-6A integration site in the Japanese population according to the literature [12]. The local alignment results showed that all TMRs from all types of HHV-6A are found in each human chromosome and at identical locations. Thus, the TMR sequence within a DR_R_ is unlikely to determine the chromosome into which HHV-6A is integrated. Furthermore, it is not possible to identify the chromosome into which the virus is integrated from the TMR sequence. However, it should be noted that this result was obtained on a small number of samples and requires further confirmation on a larger sample. It is also advisable to obtain DR_R_ sequences using nanopore sequencing technology, as this is best suited to reading DNA sequences with a high number of repeats, which is what DR_R_s are.

It is also possible that the integration site is determined by another factor or combination of factors in the HHV-6A genome. This can be determined by a comparative analysis of the HHV-6A and 6B genomes, followed by an examination of the divergent regions or genes.

If there is no structure in the HHV-6A genome that is involved in predetermining the chromosome into which the virus integrates, then the hypothesis proposed by Aswad et al. would be correct [20]. They proposed that circulating HHV-6A had lost its ability to integrate and that chromosomally integrated HHV-6A could not be released from its integrated state [20]. This conclusion is based on a phylogenetic tree constructed from 51 HHV-6A genomes. On the tree, circulating and integrated viruses diverged into two separate branches. No such pattern was observed for HHV-6B [20]. On the HHV-6B phylogenetic tree, circulating and chromosomally integrated viruses are intermixed, indicating active processes of integration, release and reintegration of HHV-6B into human chromosomes.

It follows from the above assumptions that the division of HHV-6A into clades according to integration site is based on origin from a common ancestor. It also implies that all currently observed ciHHV-6A are presumably inherited ciHHV-6A (iciHHV-6A), i.e., vertically transmitted by inheritance.

Comparative genomic analysis of ciHHV-6A and ciHHV-6B can be performed to elucidate why ciHHV-6A loses its ability to reactivate and cause reinfection. Comparative analysis of the consensus genomes of these viruses may reveal differences in gene content, gene arrangement, or other structures in the genome that may be involved in the integration/release of HHV-6A. Such discoveries will help us to better understand the mechanisms of herpes virus integration into the host genome and may allow us to find traces of integration events in the human genome.

Our result was obtained by analysing the DR_R_ only in HHV-6A, while the models of possible integration and reactivation process were developed for both HHV-6 types (6A and 6B) [18]. Confirmation of our hypothesis requires a larger statistically significant number of chromosomally integrated HHV-6A genomes from different geographical regions. Inability of the virus to reactivate may be related to the accumulated genome defects due to the long history of HHV-6A existence in the integrated form.

## Figures and Tables

**Figure 1 genes-14-00521-f001:**
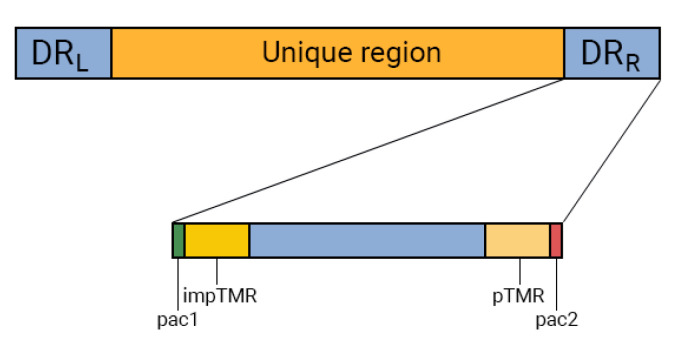
Schematic of the HHV-6A DR_R_ structure. Left and right direct repeats are marked as DR_L_ and DR_R_, respectively; impTMR – imperfect telomeric repeats; pTMR, perfect telomeric repeats; pac1 and pac2 – sequences that are involved in the packaging of the virus genome.

**Figure 2 genes-14-00521-f002:**
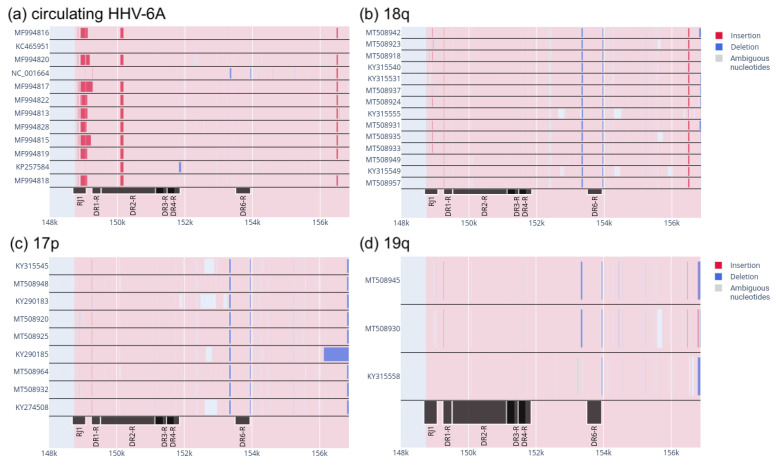
Indels in HHV-6A clades. (**a**) Indels in circulating HHV-6A. (**b**) Indels in ciHHV-6A that are presumably integrated into 18q. (**c**) Indels in ciHHV-6A that are presumably integrated into 17p. (**d**) Indels in ciHHV-6A that are presumably integrated into 19q. The area of the right DR is highlighted in pink, insertions are shown in red, deletions are shown in blue, and ambiguous nucleotides (N, any nucleotide) are shown in grey. Open reading frames in DR_R_ are shown in black below.

**Table 1 genes-14-00521-t001:** Table of viral genome classification by integration site.

Clade	NCBI ID
Circulating viruses	KJ123690, NC_001664, KP257584, MF994822, KC465951, MF994820, MF994815, MF994816, MF994817, MF994818, MF994819, MF994828, MF994813.
“17p” clade	KJ123690, KY316055, KY316048, KY290185, KY274508, KY290183, KY315545, MW049318, MW049313, MW049315, MW049316, MW049322, MK630133, MK630134, MT508920, MT508925, MT508932, MT508948, MT508964.
“18q” clade	KJ123690, KY315531, KY315540, KY315549, KY315555, MW049314, MT508918, MT508923, MT508924, MT508931, MT508933, MT508935, MT508937, MT508942, MT508949, MT508957.
“19q” clade	KJ123690, KT355575, KY316054, MG894371, KY315558, MW049317, MW049319, MW049320, MW049321, MT508930, MT508945.

**Table 2 genes-14-00521-t002:** HHV-6A genomes used for telomere repeat analysis.

Clade	NCBI ID
Circulating viruses	KJ123690, NC_001664, KP257584, MF994822, KC465951, MF994820, MF994815, MF994816, MF994817, MF994818, MF994819, MF994828, MF994813.
“17p” clade	MT508948, MT508964.
“18q” clade	MT508918, MT508923, MT508949, MT508957.

## Data Availability

The genomes used in this study are available in the public database of the Bacterial and Viral Bioinformatics Resource Centre. The GeneBank accession numbers of these genomes are listed in Appendix A.

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
