# Peer review of "The Telomeric Repeats of HHV-6A Do Not Determine the Chromosome into Which the Virus Is Integrated"

_genes, 2023, doi:10.3390/genes14020521_

Round 1

Reviewer 1 Report

The article aims to understand if the chromosomal integration of HHV-6A is determined by the telomeric repeat sequenced, for these the authors perform informatic analysis of the Direct repeat right  of circulating and chromosomally integrated HHV-6A genomes that are reported in different data bases. The authors explain that there is no pattern that can allow prediction of integration and that the sequences on the telomeric repeat do not allow the chromosomal integration. Concluding that the branches and subclades described previously are due to a common ancestor among the viruses and not to the specific integration.

I do have several points that should be improved :

1.- In the introduction and discussion th authors are referring to the DRrigth as the region to be analyzed, but the way they are referring to it is confusion, in the introduction they state that “the virus integration starts from the DRrigth region”, while in discussion they referred to Aimola et al  “showed that the integration involved DRrigth”. According to the literature DRrigth is required for integration unlike DRleft, so when they say it start from, are they referring to a possible homologous recombination as has been proposed? Or they mean that is it involved as they state in the discussion? Maybe if the keep the same concept in both parts of the paper and explain the possible mechanisms this will be clear foe any reader.

2.-The final paragraph on the introduction is confusing, I suggest to start with Aswad hypothesis, explain the alternative hypothesis and then finish with their main goal or what they are presenting.

3.-the results of the analysis of DRrigth telomeric repeats are explained in the text and presented in supplementary table S2-S6 in BED format. I am not familiar with this format, I was able to open the file with my not block, but I do not know what should I be looking for, so as non-bioinformatician and thinking on other non-experts that could read this paper if published, I suggest that a graphical representation of what they are describing in the text can be presented in a figure. I honestly cannot evaluate if the data presented is correctly analyzed, so a representation for non-experts, and a format that can be analyzed easily by anyone will be highly appreciated

4.-line 189, How is this derived from the data?

Minor point

Line 17: HHV-6A is a less well-studied virus…less studied?

Line 18: later in age….later in life?

Line 19: HHV-6A/B is transmitted…are?

Figure 1: in the legend DRL does not have the underscript.

Author Response

1. In the introduction and discussion th authors are referring to the DRrigth as the region to be analyzed, but the way they are referring to it is confusion, in the introduction they state that “the virus integration starts from the DRrigth region”, while in discussion they referred to Aimola et al  “showed that the integration involved DRrigth”. According to the literature DRrigth is required for integration unlike DRleft, so when they say it start from, are they referring to a possible homologous recombination as has been proposed? Or they mean that is it involved as they state in the discussion? Maybe if the keep the same concept in both parts of the paper and explain the possible mechanisms this will be clear foe any reader.

Response: Yes, the discussion should also include that integration starts with DRR. Formulation is changed. Also added that integration occurs by one of the putative mechanisms of homologous recombination, either break-induced replication (BIR) or single-strand annealing (SSA). (line 217)

2. The final paragraph on the introduction is confusing, I suggest to start with Aswad hypothesis, explain the alternative hypothesis and then finish with their main goal or what they are presenting.

Response: The end of the introduction has been changed. (line 53-69)

3. The results of the analysis of DRrigth telomeric repeats are explained in the text and presented in supplementary table S2-S6 in BED format. I am not familiar with this format, I was able to open the file with my not block, but I do not know what should I be looking for, so as non-bioinformatician and thinking on other non-experts that could read this paper if published, I suggest that a graphical representation of what they are describing in the text can be presented in a figure. I honestly cannot evaluate if the data presented is correctly analyzed, so a representation for non-experts, and a format that can be analyzed easily by anyone will be highly appreciated

Response: We have added a clarification on the BED format. (line 195)

4. line 189, How is this derived from the data?

Response: Explained this conclusion in more detail. (line 236)

Minor points: 

  1. Line 17: HHV-6A is a less well-studied virus…less studied?
  2. Line 18: later in age….later in life?
  3. Line 19: HHV-6A/B is transmitted…are?
  4. Figure 1: in the legend DRL does not have the underscript.

Response: Minor changes have been made.

Reviewer 2 Report

The author has investigated if any specific pattern in telomeric repeats of Human Herpes Virus-6A (HSV-6A) defines the viral integration site in the human chromosome. The author has retrieved viral sequences of HSV-6A (circulating and chromosome integrated) from public databases and analyzed them bioinformatically for insertion/deletion patterns. The study did not find any specific sequence patterns that determine the viral integration site on the chromosome but observed the affinity of telomere repeats with several human chromosomes. These findings do not fully support the conclusion as shown in the title. The author has tried to find out a correlation between telomeric repeat sequences and viral integration sites bioinformatically but does not perform any mechanistic study (wet lab) that confirms the role of tandem repeats in determining chromosomal integration sites.  Besides, there are several other concerns that are listed below.

Ø  The title is misleading and not very clear. I would suggest changing the title to either something conclusive according to the data shown or it might be something such as The chromosomal integration of HHV-6A cannot be determined merely on telomeric repeat sequences”.

Ø  It is not clear whether the strains mentioned in table 1 are reference strains for different viral clades or whether the studied strains are classified into clades by analyzing sequences in context to reference. Please clarify and explain clearly in the text. If these are the studied strains, they can be represented as a phylogenetic tree for better understanding.

Ø  The results are not very understandable. I would suggest providing a pictorial outline of the results by mentioning each step from strains/sequence selection (How many strains are included or excluded in screening; how many were circulating/integrated strains) to the classification into clades/identifying indels.

Ø  It is mentioned in the results that there are no patterns of insertion and deletion distribution that would be characteristic only for chromosomally integrated HHV-6A.” But I see the pattern of deletions in figure 2 (two blue lanes shown at 154K positions) that is completely missing in circulating strains (panel a) as compared to integrated strains, ciHHV-6A (panels b, c & d). Similarly, it is marked that There are also no patterns of indels that would allow us to distinguish viruses of different integration sites from each other.” However, I see the one lane of insertion (red) that is present only in 18q clade (panel b) and missing in other clades (panels c, d & a). If I understand correctly, these observations show that there is a pattern of indels among different clades. Please explain and if it is correct then update these statements; the conclusion/title might be changed accordingly.

Ø  The author has identified indels in telomeric repeats as shown in figure 2. I am wondering if these indels were also identified in the pac1/pac2 regions of the right direct repeats (DRR) because these regions determine the disintegration of the viral genome from chromosomes and any specific pattern in these regions might be specific to each viral integration clade.

Ø  Include the concluding remarks at the end of the discussion.

Author Response

1. The title is misleading and not very clear. I would suggest changing the title to either something conclusive according to the data shown or it might be something such as “The chromosomal integration of HHV-6A cannot be determined merely on telomeric repeat sequences”.

Response: New title suggestion is ”The telomeric repeats of HHV-6A do not determine the chromosome into which the virus is integrated”.

2. It is not clear whether the strains mentioned in table 1 are reference strains for different viral clades or whether the studied strains are classified into clades by analyzing sequences in context to reference. Please clarify and explain clearly in the text. If these are the studied strains, they can be represented as a phylogenetic tree for better understanding.

Response: There are only 2 reference genomes for HHV-6A, they are NC_001664 and KJ123690. The first is 159378 bp long and was isolated in Uganda in 1994. The second is 156864 bp long and was isolated in the USA in 2013. We decided to use KJ123690 as a reference because the vast majority of the genomes used in the study were obtained later than 2013.

3. The results are not very understandable. I would suggest providing a pictorial outline of the results by mentioning each step from strains/sequence selection (How many strains are included or excluded in screening; how many were circulating/integrated strains) to the classification into clades/identifying indels.

Response: This information is presented in the relevant results sections. Table 2 has been added to the TMR analysis section to clarify how many and which genomes were used. (line 185)

4. It is mentioned in the results that “there are no patterns of insertion and deletion distribution that would be characteristic only for chromosomally integrated HHV-6A.” But I see the pattern of deletions in figure 2 (two blue lanes shown at 154K positions) that is completely missing in circulating strains (panel a) as compared to integrated strains, ciHHV-6A (panels b, c & d). Similarly, it is marked that “There are also no patterns of indels that would allow us to distinguish viruses of different integration sites from each other.” However, I see the one lane of insertion (red) that is present only in 18q clade (panel b) and missing in other clades (panels c, d & a). If I understand correctly, these observations show that there is a pattern of indels among different clades. Please explain and if it is correct then update these statements; the conclusion/title might be changed accordingly.

Response: 2 deletion sites in the 154 kbs region are observed in one circulating HHV-6A (NCBI ID: NC_001664). Therefore, we cannot say that these deletions are not present in the circulating ones. Also, the left of the two blue lines (deletions) is missing in the KY315558 genome from the virus group integrated into chromosome 19 (panel d), which does not allow us to say that a pair of these deletions are present in all chromosomally integrated HHV-6A.

In panel b of genome KY31555, the insertion occurs in an incompletely sequenced region (marked in grey), so it is impossible to say for sure whether this insertion is of the same length as those in other genomes in this clade.

In any case, a large sample of genomes is needed to say with certainty whether deletion/insertion patterns are present for any of the clades. (line 151-172)

5. The author has identified indels in telomeric repeats as shown in figure 2. I am wondering if these indels were also identified in the pac1/pac2 regions of the right direct repeats (DRR) because these regions determine the disintegration of the viral genome from chromosomes and any specific pattern in these regions might be specific to each viral integration clade.

Response: pac1/pac2 are challenging to study because they are adjacent to human telomeres. Such regions are difficult to sequence using Illumina's short reads and require the use of the more expensive long read MinION technology. Unfortunately, the HHV-6A genomes we collected from the BV-BRC database were sequenced by Illumina and do not have pac1/pac2 at the ends (except for circulating HHV-6A). To investigate this issue, a special study should be carried out to obtain these sequences.

6. Include the concluding remarks at the end of the discussion.

Response: The conclusions have been updated.

Reviewer 3 Report

In this manuscript, the authors investigate if sequence patterns (emphasis on indels) in the DRR  of HHV6-A have predictive value on the chromosome in which the virus is inserted. 

Comments:

 1.      The authors investigate only the DRR, and within it only specific sections (telomere repeats) based on a reference from an in vitro study. However, the current consensus is still that ici-HHV-6A is integrated using both direct repeats. Any reason for not analyzing DRl ?

2.      Lines 61-63 are not clear. Please refine the point and sentence structure. Any reason to believe that currently circulating HHV6-A cannot integrate? Any differences in e.g. telomeres between circulating and integrated forms that would support this?

3.      In line 80 the authors state that SNPs were not analyzed. Could the authors explain why?

4.      Among the 66 HHV-6A genomes gathered, have all the telomere areas been gap-filled by PCR sequencing, and or are there sequences based solely on high-throughput short-read sequencing (such as Illumina)? The reason for this question is that long telomere repeat areas are practically impossible to assemble reliably by short-read sequencing alone. Thus, the reliability of these sequences is poor. 

5.      The authors base their analysis on available sequences of known integration sites. However, for most of the sequences used  (Suppl. table 1) the chromosome in which the integration occurs appears as predicted. Are the confirmed integrations done by FISH or similar? How have the predictions been done? How reliable are them? 

6.      Can the authors specify which references (n=19) were used for the analysis of the DRR telomeric repeats?

 7.      The authors find the telomere repeat sequences in ciHHV-6A to be identical to those of chromosomes 11, 17, 18, 19, and 22. Any reason for not studying other chromosomes? Since the telomere repeats in HHV-6A are identical to the telomere repeats of humans, the integration could likely happen in all the other chromosomes.

Author Response

1. The authors investigate only the DRR, and within it only specific sections (telomere repeats) based on a reference from an in vitro study. However, the current consensus is still that ici-HHV-6A is integrated using both direct repeats. Any reason for not analyzing DRl ?

Response: DRR and DRL have identical sequences within the same virus. The only difference is the orientation of the direct repeat region. This is probably the reason why viral integration only starts in a specific orientation, starting from the DRR. If complementation has occurred for a perfect TMR within a DRR, it will also occur for an imperfect TMR within a DRL. Therefore, we focused only on the DRR analysis.

2. Lines 61-63 are not clear. Please refine the point and sentence structure. Any reason to believe that currently circulating HHV6-A cannot integrate? Any differences in e.g. telomeres between circulating and integrated forms that would support this?

Response: The conclusion that chromosomal integration of circulating HHV-6A is impossible is based on the phylogenetic tree. If such cases were observed, circulating and integrated viruses would be mixed, as observed for HHV-6B in Aswad et al. (line 53-62). To confirm this hypothesis, a large-scale study aimed at identifying both circulating and integrated viruses, followed by the phylogenetic tree, is certainly needed.

Mutations in the perfect telomeric repeats (pTMR) of circulating HHV-6A or in pac1/pac2 of integrated HHV-6A could explain the reduced ability or complete inability to integrate into human chromosomes.

3. In line 80 the authors state that SNPs were not analyzed. Could the authors explain why?

Response: Insertions and deletions affect longer regions than SNPs and are more likely to alter the regulatory function of the direct repeat region. Of course, single nucleotide substitutions can accumulate over time and also alter DRR function. Studying such changes was partly done when analysing the affinity of viral telomeric repeats to human telomeric repeats because it also considers SNPs.

The study of SNPs for the whole right direct repeat region requires a slightly different approach to that used in this study and may be the subject of the next paper.

4. Among the 66 HHV-6A genomes gathered, have all the telomere areas been gap-filled by PCR sequencing, and or are there sequences based solely on high-throughput short-read sequencing (such as Illumina)? The reason for this question is that long telomere repeat areas are practically impossible to assemble reliably by short-read sequencing alone. Thus, the reliability of these sequences is poor.

Response: The tandem repeats that make up telomere repeats are not perfect; they have single nucleotide substitutions. Therefore, with sufficient coverage, it is likely that short Illumina reads can be assembled. For TMR analysis in this study, we pre-filtered genomes where the forward repeat region was either poorly assembled or missing. The Tandem Repeats Finder result confirms that telomeric repeats are present in the remaining genomes.

5. The authors base their analysis on available sequences of known integration sites. However, for most of the sequences used  (Suppl. table 1) the chromosome in which the integration occurs appears as predicted. Are the confirmed integrations done by FISH or similar? How have the predictions been done? How reliable are them?

Response: Proven integration sites were determined by FISH. The prediction of the integration site was made in Aswad et al. 2021 based on a phylogenetic tree. They found that viruses with proven integration sites (FISH method) were distributed in separate clades. Based on this, it was assumed that viruses from the respective clades had corresponding integration sites.

6. Can the authors specify which references (n=19) were used for the analysis of the DRR telomeric repeats?

Response: Table with NCBI ID added to the article (Table 2). (line 185)

7. The authors find the telomere repeat sequences in ciHHV-6A to be identical to those of chromosomes 11, 17, 18, 19, and 22. Any reason for not studying other chromosomes? Since the telomere repeats in HHV-6A are identical to the telomere repeats of humans, the integration could likely happen in all the other chromosomes.

Response: Yes, cases of virus integration into 1q, 6q, 9q, 10q, 11p, 17p, 18p, 19q, 22q and Xp chromosomes have been described in the literature. However, some of these studies do not specify the type of virus (6A or 6B). We have used the chromosomes that are most characteristic of 6A. If HHV-6A does not lose its ability to integrate, it is likely to be observed in all human chromosomes.

Round 2

Reviewer 1 Report

The authors have answerd all my concerns

Author Response

Thank you very much for the review, your comments were very valuable and helped to improve the article!

Reviewer 2 Report

The author has revised the article by addressing most of the concerns. I think it can be considered for submission. 

Author Response

Thank you very much for your comments, they really helped to improve the article!

Reviewer 3 Report

I thank the authors for addressing my questions and for the clarifications made in the manuscript file.

As minor revisions, please correct line 151 "two distinct deletions AT". Same in line 169 "an insertion in the region AT about".

Author Response

1. As minor revisions, please correct line 151 "two distinct deletions AT". Same in line 169 "an insertion in the region AT about".

Response: minor changes have been made

Thank you very much for your comments, they were very helpful in finalising the article!